# Ranking evaluation metrics from a group-theoretic perspective

## Abstract

Searching for always better-performing machine learning techniques requires continuously comparing with well-established methods. While facing the challenges of finding the right evaluation metric to prove the strengths of the proposed models, choosing one metric despite another might hide the method's weaknesses intentionally or not. Conversely, one metric fitting all applications is probably not existing and represents a hopeless search.

In several applications, comparing rankings represents a severe challenge: various metrics strictly correlated to the context appeared to evaluate their similarities and differences. However, most metrics spread to other areas, although a complete understanding of their internal functioning is often missing, leading to unexpected results and misuses. Furthermore, as distinguished metrics focus on different aspects and rankings' characteristics, the comparisons of the models' results outputs given by the various metrics are often contradicting.

We propose to theorize rankings using the mathematical formality of symmetric groups to rise above the possible contextualization of the evaluation metrics. We prove that contradictory evaluations frequently appear among pairs of metrics, introduce the *agreement ratio* to measure the frequency of such disagreement, and formally define essential mathematical properties for ranking evaluation metrics. We finally check if any of these metrics is a *distance* in the mathematical sense. In conclusion, our analysis underlines the inconsistencies' reasons, compares the metrics purely based on mathematical concepts, and allows for a more conscious choice based on specific exigencies.

## 1 Introduction

Evaluating methods is essential in any machine learning field; however, finding the *right* evaluation metric assessing one method's strengths without providing unfair comparisons to others is not always straightforward. The evaluation of methods whose results are rankings is generally a great challenge. Among these methods, Recommender Systems (RS) have become a prosperous research area since the mid-1990s. The recommendation algorithms output lists of recommended items Adomavicius & Tuzhilin (2005), similar to Information Retrieval (IR) techniques, that look for relevant information in huge search spaces given a specific information quest Schütze et al. (2008). Other methods also provide rankings as outputs: In feature ranking and selection approaches, features are ordered according to their usefulness in the task at hand Khaire & Dhanalakshmi (2022); Rank and fair rank aggregation aim to obtain unique rankings given a set of (possibly biased) rankings. The evaluation of all these methods often includes comparing rankings.

Many context-specific evaluation metrics are available, particularly for evaluating RS. The same metrics spread in the other evaluation contexts, i.e., feature selection and rank aggregation. For IR and RS techniques, it became evident that comparing methods is a significant challenge, and contradictory evaluations are at the order of the day. Offline metrics for RS compare the output of the algorithms with external ground truth rankings and are easily applicable externally to RS Cañamares et al. (2020); Beel & Langer (2015). Together with Information Retrieval evaluation measures (e.g., $DCG$), offline metrics comprehend measures of errors and relevance-based metrics. Many offline metrics spread to other machine learning areas to compare rankings; examples are recall@$k$ and NDCG, both used in feature selection approaches. Choosing evaluation

metrics to compare two rankings is often non-straightforward, and the many inconsistencies among the produced evaluations hinder their credibility. Furthermore, validating ranking metrics experimentally is typically unfeasible and does not allow for good generalization in other experimental setups.

Our paper proposes a list of desirable theoretical properties for ranking evaluation metrics and provides the mathematical background for each. We first transfer the problem of comparing rankings to symmetric groups. By generalizing to symmetric groups $S_n$, we detach from specific machine learning contexts; our goal is finding an answer to the question *which mathematical properties are essential in the evaluations?* rather than *what is the metric of success?* for a specific exigence. Symmetric groups are the most general mathematical structure on which we could represent rankings, thus explaining our choice. Our approach interprets ranking evaluation metrics as functions defined over a mathematical group, thus allowing for a theoretical analysis of the mathematical properties satisfied by the metrics. We provide insights and an understanding of the use and the goals optimized by the ranking evaluation metrics. Eventually, this allows for a conscious choice of evaluation metrics to measure the similarity among rankings in specific contexts.

Our work provides a theoretical framework and we detach from specific contexts of application. As a matter of fact, the current literature appears to be rather limited to considering only specific applications (for an exception, see Diaconis, 1988). In Section 4 we provide motivation examples, introducing the notion of *inconsistency* among metrics and the agreement ratio; additionally, we use the theoretical definitions of the metrics to cluster them in Section 5.1. Section 6 describes desirable well-founded mathematical properties for ranking evaluation metrics; Table 2 summarizes which properties are satisfied by the various metrics. We claim that, in our definitions, none of the metrics is a mathematical distance and modify the *discounted cumulative gain* to obtain one. Finally, Section 7 explores the relationships among the various properties. Although jumping in the generalization offered by symmetric groups, we aim to do not forget the contexts in which the metrics have been developed, and highlight in which contexts the mentioned properties are particularly desirable.

## 2 Related work

The literature on ranking evaluation metrics is vast and extensive for RS evaluation. Several works investigated how reliable offline and online evaluation metrics are and how they relate to each other within RS evaluation (Valcarce et al., 2018; Liu & Yu, 2021; Gunawardana et al., 2012; Silveira et al., 2019; Li et al., 2011). Herlocker et al. (2004) surveyed most evaluation metrics used for comparing collaborative filtering RS and proposed a theoretical division of the metrics. Liu et al. (2009) precisely describe most of the metrics typically used for RS and IR techniques; however, this work concentrates on metrics and algorithms specifically built for these applications which limits the transfer to different contexts. Järvelin & Kekäläinen (2002) presented various metrics based on cumulative gain, pointing out their main advantages and drawbacks. The work by Hoyt et al. (2022) proposes a theoretical foundation for rank-based evaluation metrics, particularly considering the metrics *hits at k*, *mean rank* and *mean reciprocal rank* MRR and they defined some desiderata for link prediction in knowledge graphs. Amigó et al. (2018) define a set of properties for IR metrics and show that none of the existing ones satisfy all the properties proposed. Other works focus on metrics for RS and their intrinsic properties, e.g., Buckley & Voorhees (2004) and Valcarce et al. (2020) performed a comparison of ranking metrics for the top-$n$ recommendations, in particular being interested in items and users missing at random, in the robustness to incompleteness and the discriminating power of each of the metrics. Another question is whether ranking evaluation metrics are *interval scales*; Ferrante et al. (2018) explored the scale properties of IR metrics analyzing both binary and non-binary relevance, set-based and rank-based evaluation metrics. Furthermore, real-world applications such as the design of strategies based on customers' feedback, experts' opinion analysis, and allocation of priorities in R&D extended the interest in defining distances among rankings in Dwork et al. (2001); Sculley (2007); Kim et al. (2013); the focus of the problem statement is *rank aggregation* to find representatives for communities of voters. As an example among similarly scoped works, we find Cook et al. (1986); Fligner & Verducci (1986). Hassanzadeh & Milenkovic (2014) insisted on defining distances for rankings based on similarity. The work by Diaconis (1988) is worth special attention. The author focuses on six metrics on symmetric groups, among them *Kendall's τ* and *Spearmann's ρ* while the other considered metrics are rather uncommon in machine learning. The author studies them from a statistical perspective and analyzes their theoretical properties. Some properties that

we define in our work present strong similarities to some in Diaconis (1988). Among the defined properties, we find the *interpretability* or whether the metrics measure something humanly tangible; the *tractability*, i.e., the so-called computational complexity in computer science; the *sensitivity* defined as the ability of one metric to range among the available counter-domain; the *theoretical availability* that asks whether a metric is studied and used enough in the state-of-the-art works. We add to its work considering an extensive set of ranking evaluation metrics and defining additional theoretical mathematical properties. Furthermore, we contextualize these metrics to the specific contexts, emphasizing in which application contexts their properties are mostly desirable.

Choosing proper and fair evaluation metrics is a fast-growing field in computer science. Some of the cited ranking evaluation metrics have been harshly criticized for their comparisons' reliability in the evaluations (Tamm et al., 2021). The central gap to be spotted in the literature is the complete silence concerning the use of standard RS ranking evaluation metrics in other contexts. In other areas, works defining properties for metrics are popping out in the state-of-the-art literature, e.g., Gösgens et al. (2021a;b). Furthermore, also older literature offers some works focusing on analyzing the influence of the metric in determining (sub)optimal models for supervised learning tasks, e.g., in Caruana & Niculescu-Mizil (2004), where several metrics have been analyzed theoretically in a differential geometry perspective. We structured the paper based on a successful strategy of defining mathematical properties for ranking evaluation metrics, each justified from a mathematical point of view; the generalization to rankings on symmetric groups allows us to rise above the limitations of the literature on RS metrics and achieve a context-independent analysis applicable for rankings appearing in any machine learning method.

## 3 Ranking evaluation metrics

Most RS evaluation metrics can be used to compare rankings of $n$ elements, except the ones requiring additional context-specific information and the *online evaluation metrics*. Among widely spread metrics, such as DCG, *recall*, or MSE, various less-known metrics are used in the literature when comparing rankings. We report the considered evaluation metrics list in Table 1 and refer to Appendix C for the formal definitions. We distinguish among *ranking aware metrics*, aware of the position in the ranking of single items, and *flat metrics* not considering the position in the ranking of the items; in the second grouping, we find two subcategories: set based metrics and the ones assigning equal importance to each position in the ranking (see Figure 1 (b)). Furthermore, we cluster ranking evaluation metrics from a theoretical point of view into four main groups: confusion matrix, correlation, error, and cumulative gain. The *confusion matrix based CMB metrics* are based on the number of correctly retrieved elements, elements incorrectly classified, and correctly non-retrieved items. They are essentially set-based metrics. The *correlation based metrics* are statistics measuring the ordinal association between two measured quantities. NDPM is slightly differently defined, although it satisfies the same characteristics. Often used to analyze the performance of predicting models, *error based metrics* compute the difference between the true and predicted values. Their evaluation, however, does not depend on which are the predicted and the true labels as their internal computation involve either squared or absolute value computation. They are often used for comparing rankings or scores; we will consider here the metrics when used to compare two rankings, independently from the presence or not of a ground truth ranking. They can be classified as flat metrics. Finally, *cumulative gain based metrics* focus on the rankings of the single elements.

Given a finite set $\mathcal{N} = \{1, \ldots, n\}$, we call *symmetric group* $S_n$ the set of bijective functions from $\mathcal{N}$ to $\mathcal{N}$; $S_n$ is a group with respect to the *function composition* as group operation. Note that the only possible bijective functions from a finite group to itself are the permutations over the elements in $\mathcal{N}$, and the size of $S_n$ is $n!$. We indicate permutations using greek letters, i.e., $\sigma \in S_n$, and the *identity function* id is the identity function, i.e., $\text{id} : i \mapsto \text{id}(i) = i$ for all $i \in \{1, \ldots, n\}$. If there are no chances of confusion, we do not indicate the length of the rankings. Given $\sigma \in S_n$, $\sigma(i)$ indicates the position in which the $i$th element is sent by $\sigma$; $\sigma_{|t} = (\sigma(1), \ldots, \sigma(t))$ indicates the ranking of the first $t$ elements while $\text{set}(\sigma_{|t})$ is the set of the first $t$ elements ranked regardless the ordering. Given $\sigma, \nu \in S_n$, $\sigma \circ \nu \in S_n$ is the permutation defined by $\sigma \circ \nu(i) = \sigma(\nu(i))$ for all $i \in \{1, \ldots, n\}$. The composition of permutations is not commutative, i.e., generally $\sigma \circ \nu \neq \nu \circ \sigma$. The cycle decomposition theorem states that each permutation can be rewritten in a unique way as the composition of relatively disjoint permutations (or *cycles*). Finally, we call a *(single) swap* a

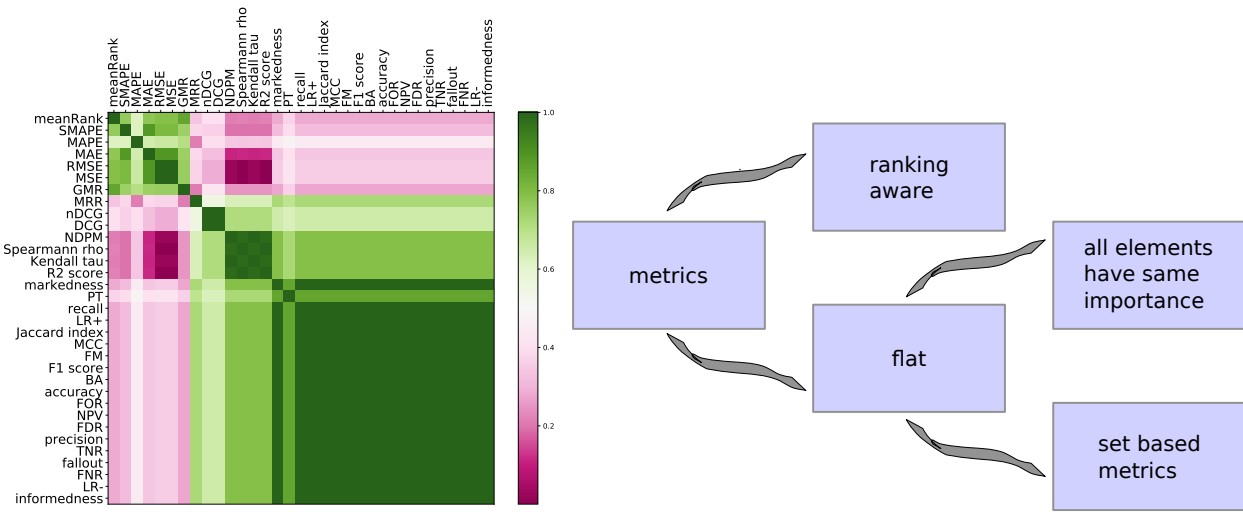

Figure 1: **On the left:** Heatmap of the disagreement ratios among pairs of ranking evaluation metrics. **On the right:** The theoretical subdivision of the metrics.

| | |
|---|---|
| ranking aware metrics | **nDCG**, **DCG**, meanRank, GMR, MRR |
| equal importance | *SMAPE*, *MAPE*, *MAE*, *RMSE*, *MSE*, $R^2$ *score*, Spearmann $\rho$, Kendall's $\tau$, NDPM |
| set based metrics | markedness, PT, recall, LR+, Jaccard index, F1 score, accuracy, FDR, precision, TNR, fallout, FNR, LR- informedness, NPV, FOR, BA, FM, MCC |

Table 1: List of metrics analyzed grouped according to their definitions and properties; bold, italic, underlined, and plain text indicate **cumulative gain**, *error*, CMB, and correlation based metrics. MRR, GMR and meanRank do not fall into any of the groups and are blue color-coded.

permutations $\sigma = (j\ k) \in S_n$ swapping only the two elements $j, k$ in $\mathcal{N}$, i.e., $\sigma(i) = j$ if $i = k$, $\sigma(i) = k$ if $i = j$ and $\sigma(i) = i$ if $i \neq j, k$. Swaps can be easily find in the literature, also under different names; as an example, Hassanzadeh & Milenkovic (2014) refers to them as *transpositions*. The set of swaps over $\mathcal{N}$ is not *closed* with respect to the group operation, i.e., composing two swaps we possibly optain a permutation that is not a swap..

## 4 Motivational example

Having a clear and sufficient understanding of the theoretical fundamentals of the metrics is essential to choose metrics for evaluating newly proposed methods and comparing them with existing ones. An appropriately chosen metric might improve the attractiveness of a newly proposed method, but it can also cover up the methods' drawbacks. A deeper understanding of the used metrics hopefully allows for fairer and more reproducible results. Generally, a *ranking evaluation metric* is a function $m : S_n \times S_n \to \mathbb{R}_+$; In some cases, we deal with metrics that take as input only one ranking and compare the ranking in question against an underlying *optimal* one.

**Definition 1.** *Two ranking evaluation metrics* $m_1, m_2$ *are* non-consistent *(or* inconsistent*) if* $\exists \sigma, \mu, \nu \in S_n$ *such that*

$$\begin{aligned} m_1(id, \sigma) \leq m_1(id, \mu) \ &\wedge \ m_2(id, \sigma) \leq m_2(id, \mu) \\ m_1(id, \sigma) \leq m_1(id, \nu) \ &\wedge \ m_2(id, \sigma) > m_2(id, \nu) \end{aligned} \tag{1}$$

*If for each choice of* $\sigma, \mu, \nu \in S_n$ *equation 1 is not satisfied, we say that* $m_1, m_2$ *are* consistent.

To say that two metrics are non-consistent , it is enough to find three rankings such that $\sigma$ is the closest to the trivial ranking according to $m_1$ but not according to $m_2$. $m_1$ and $m_2$ are consistent, if $\forall \sigma, \mu \in S_n$, equation 1

is not satisfied with respect to id. The first line of equation 1 guarantees that the $\tilde{m}_2 = -m_2$ is still non-consistent with $m_1$ avoiding the case of $m_1$ and $-m_2$ being consistent. Pairs of metrics theoretically similar are not necessarily *consistent* with each other; Most ranking evaluation metrics' pairs exhibit *inconsistencies*. Method A can appear better than Method B using one metric and worse according to a different metric. We give additional details in see Section 5.1. Furthermore, we will give a sufficient condition under which evaluation metrics do not allow for inconsistencies.

## 5 Ranking measures fundamentals

Most methods returning rankings of items are evaluated by comparing the output with the ground truth or the desired output. We can theorize the ranking evaluation metrics as functions over the symmetric group $S_n$; their aim is quantifying the differences between two rankings. Each metric considers different rankings aspects: nDCG assumes that highly relevant documents are most useful when appearing earlier in the ranking and that highly relevant documents are more useful than marginally relevant documents, which are, in turn, more useful than non-relevant documents; *Kendall's $\tau$ score* measures the smallest number of swaps of adjacent elements that transform one ranking into the other (see Kendall (1948)); *Precision* needs an additional parameter $k$ to compute the set of retrieved elements, and its definition relies on the confusion matrix. Such distinctions allow clustering evaluation measures from a theoretical perspective, as seen in Section 3. Moreover, constructing examples of inconsistencies among evaluation measures is often trivial.

In the following sections, we refer to the metrics that consider only the first $k$ ranked elements as metrics@$k$; confusion matrix based metrics are an example. Almost any metric can be reduced to a metric@$k$ considering only the first $k$ elements ranked; However, throughout the paper, we will always consider metrics evaluating the full rankings unless differently specified. Some metrics require the set of relevant elements to be contained in the set of retrieved elements, e.g., the MRR, meanRank and GMR.

### 5.1 Clustering by agreement

We quantify the frequency under which inconsistencies among pairs of metrics pop up in the evaluation of different-sized rankings introducing a coefficient of agreement among two metrics:

**Definition 2.** *For any $\sigma \in S_n$ fixed, the $\sigma$ agreement ratio among two ranking evaluation metrics $m_1, m_2$ is*

$$AR_{m_1,m_2}^{\sigma} = \frac{1}{|\mathcal{T}|(|\mathcal{T}| - 1)} \sum_{\mu,\nu \in \mathcal{T}} f_{\sigma}^{m_1,m_2}(\nu,\mu)$$

*where $\mathcal{T} \subseteq S_n$, $f_{\sigma}^{m_1,m_2}(\nu,\mu) = \mathbb{1}\{\mu,\nu$ are consistent w.r.t. $\sigma\}$ and $\mathbb{1}$ is the indicator function, i.e., equals 1 in the case the argument is satisfied and 0 otherwise.*

The agreement ratio measures how many inconsistencies exist among two evaluation metrics on a subset of $\mathcal{P}(S_n)$ and equals 1 whenever $m_1$ and $m_2$ are consistent. Up to renaming the elements in $\mathcal{N}$, we suppose that $\sigma = $ id such that the agreement ratio does not depend on $\sigma$. Valcarce et al. (2020) proposed directly studying the correlation among the evaluation metrics evaluations for RS using Kendall's $\tau$ score. However, as we included Kendall's $\tau$ score in our analysis, we preferred introducing a non-circular evaluation of the disagreements' frequency.

Figure 1 (left) shows that the evaluation metrics similar from a theoretical point of view have a high agreement ratio (green color); The plot refers to $\mathcal{T}$ being a random subset sample of $S_{100}$, containing 10000 random pairs of rankings. For reasons of symmetry (see Section 6.2), we only considered an equal number of retrieved and relevant elements and fixed it to 30. In the case of no agreements for most considered pairs of rankings among $m_1$ and $m_2$, we considered the metric $-m_2$, which leads to a reversed agreement ratio with $m_1$. The agreement ratio being symmetric, the upper triangle of the heatmap is sufficient for the analysis. The green color represents pairs of metrics essentially agreeing, while the pink color represents high disagreement among pairs of metrics; finally, the white color represents a partial agreement. The agreement ratio is evenly distributed among the metrics; the number of highly agreeing pairs of metrics is not significantly different from the number of pairs highly disagreeing. Similar results are obtained by varying the length $n$ of the rankings and the number of relevant elements we are interested in retrieving.

| | recall | FNR | fallout | TNR | precision | FDR | NPV | FOR | accuracy | BA | F1 score | FM | MCC | Jaccard index | markedness | LR- | informedness | PT | LR+ | MSE | RMSE | MAE | MAPE | SMAPE | R² score | Kendall's τ | Spearmann ρ | NDPM | DCG | nDCG | MRR | GMR | meanRank |
|---|---|---|---|---|---|---|---|---|---|---|---|---|---|---|---|---|---|---|---|---|---|---|---|---|---|---|---|---|---|---|---|---|---|
| id. indisc. | ✗ | ✗ | ✗ | ✗ | ✗ | ✗ | ✗ | ✗ | ✗ | ✗ | ✗ | ✗ | ✗ | ✗ | ✗ | ✗ | ✗ | ✗ | ✗ | ✗ | ✗ | ✗ | ✗ | ✗ | ✗ | ✗ | ✗ | ✗ | ✓ | ✓ | ✗ | ✗ | ✗ |
| symmetry | ✓ | ✓ | ✓ | ✓ | ✓ | ✓ | ✓ | ✓ | ✓ | ✓ | ✓ | ✓ | ✓ | ✓ | ✓ | ✓ | ✓ | ✓ | ✓ | ✓ | ✓ | ✗ | ✓ | ✓ | ✓ | ✓ | ✓ | ✓ | ✗ | ✗ | ✗ | ✗ | ✗ |
| rob. I(a) 10 | 0.07 | 0.07 | 0.03 | 0.03 | 0.07 | 0.07 | 0.03 | 0.03 | 0.04 | 0.04 | 0.07 | 0.07 | 0.10 | 0.06 | 0.10 | 0.18 | 0.10 | 0.06 | 0.40 | 2.70 | 0.35 | 0.33 | 13.46 | 6.01 | 0.33 | 0.12 | 0.16 | 0.06 | 0.63 | 0.02 | 0.01 | 0.64 | 0.61 |
| rob. I(a) 50 | 0.01 | 0.01 | 0.00 | 0.00 | 0.01 | 0.01 | 0.00 | 0.00 | 0.00 | 0.00 | 0.01 | 0.01 | 0.01 | 0.00 | 0.01 | 0.02 | 0.01 | 0.02 | 0.08 | 11.61 | 0.29 | 0.28 | 4.17 | 1.05 | 0.06 | 0.03 | 0.03 | 0.00 | 1.66 | 0.00 | 0.00 | 0.56 | 0.53 |
| rob. I(a) 100 | 0.00 | 0.00 | 0.00 | 0.00 | 0.00 | 0.00 | 0.00 | 0.00 | 0.00 | 0.00 | 0.00 | 0.00 | 0.00 | 0.00 | 0.00 | 0.00 | 0.00 | 0.02 | 0.03 | 23.19 | 0.31 | 0.29 | 2.54 | 0.56 | 0.03 | 0.00 | 0.01 | 0.00 | 2.19 | 0.00 | 0.00 | 0.53 | 0.50 |
| rob. I(b) 10 | 0.16 | 0.16 | 0.07 | 0.07 | 0.16 | 0.16 | 0.07 | 0.07 | 0.08 | 0.08 | 0.16 | 0.16 | 0.22 | 0.13 | 0.22 | 0.38 | 0.22 | 0.15 | 0.83 | 6.85 | 0.88 | 0.87 | 39.52 | 15.46 | 0.83 | 0.31 | 0.42 | 0.11 | 1.19 | 0.04 | 0.03 | 1.75 | 1.68 |
| rob. I(b) 50 | 0.03 | 0.03 | 0.01 | 0.01 | 0.03 | 0.03 | 0.01 | 0.01 | 0.01 | 0.01 | 0.03 | 0.03 | 0.04 | 0.02 | 0.04 | 0.06 | 0.04 | 0.06 | 0.31 | 68.52 | 1.69 | 1.72 | 38.92 | 6.29 | 0.33 | 0.11 | 0.16 | 0.02 | 3.44 | 0.00 | 0.00 | 2.77 | 2.61 |
| rob. I(b) 100 | 0.01 | 0.01 | 0.00 | 0.00 | 0.01 | 0.01 | 0.00 | 0.00 | 0.00 | 0.00 | 0.01 | 0.01 | 0.02 | 0.01 | 0.02 | 0.03 | 0.02 | 0.04 | 0.18 | 190.59 | 2.35 | 2.42 | 40.46 | 4.47 | 0.23 | 0.08 | 0.11 | 0.01 | 5.28 | 0.00 | 0.00 | 3.23 | 3.05 |
| rob II | ✗ | ✗ | ✗ | ✗ | ✗ | ✗ | ✗ | ✗ | ✗ | ✗ | ✗ | ✗ | ✗ | ✗ | ✗ | ✗ | ✗ | ✗ | ✗ | ✓ | ✓ | ✓ | ✓ | ✗ | ✓ | ✓ | ✓ | ✗ | ✗ | ✗ | ✗ | ✗ | ✗ |
| WSD | ✓ | ✓ | ✓ | ✓ | ✓ | ✓ | ✓ | ✓ | ✓ | ✓ | ✓ | ✓ | ✓ | ✓ | ✓ | ✓ | ✓ | ✓ | ✓ | ✓ | ✓ | ✓ | ✓ | ✓ | ✓ | ✗ | ✗ | ✗ | ✓ | ✓ | ✓ | ✓ | ✓ |
| sensitivity | ✗ | ✗ | ✗ | ✗ | ✗ | ✗ | ✗ | ✗ | ✗ | ✗ | ✗ | ✗ | ✗ | ✗ | ✗ | ✗ | ✗ | ✗ | ✗ | ✗ | ✗ | ✗ | ✓ | ✓ | ✗ | ✗ | ✗ | ✓ | ✗ | ✗ | ✓ | ✓ | ✓ |
| stability | ✓ | ✓ | ✗ | ✗ | ✓ | ✓ | ✗ | ✗ | ✗ | ✓ | ✓ | ✓ | ✗ | ✓ | ✗ | ✓ | ✓ | ✓ | ✓ | ✓ | ✓ | ✓ | ✓ | ✓ | ✓ | ✓ | ✓ | ✓ | ✓ | ✓ | ✓ | ✗ | ✗ |
| distance | ✗ | ✗ | ✗ | ✗ | ✗ | ✗ | ✗ | ✗ | ✗ | ✗ | ✗ | ✗ | ✗ | ✗ | ✗ | ✗ | ✗ | ✗ | ✗ | ✗ | ✗ | ✗ | ✗ | ✗ | ✗ | ✗ | ✗ | ✗ | ✓ | ✓ | ✗ | ✗ | ✗ |

Table 2: Summary table of the property and the metrics that satisfy them. Type I Robustness property: average of the absolute differences from equation 5 and equation 6; in green, the ones < 0.05.

# 6 Ranking evaluation metrics' properties

Intending to add clarity over the metrics used for rankings in various contexts, we define mathematical properties, give insights on whether they are satisfied by the metrics, and prove our theoretical claims. We summarize the findings in Table 2, the code will be on Github upon acceptance[1]. We define the *linear equivalence* for ranking evaluation metrics as follows:

**Definition 3.** *Two metrics $m_1$ and $m_2$ are* linearly equivalent *($m_1 \sim m_2$) if there exists a non-constant linear function $f$ such that either $f(m_1(\sigma, \nu)) = m_2(\sigma, \nu)$ or $m_1(\sigma, \nu) = f(m_2(\sigma, \nu))$ for any $\sigma, \nu \in S_n$.*

A *linear equivalence* is an equivalence relation on the space of ranking evaluation metrics, i.e., it satisfies (a) the *reflexive property*, i.e., $m \sim m$ where $f$ is the identity function; (b) the *symmetry property*, i.e., $m_1 \sim m_2 \leftrightarrow m_2 \sim m_1$ (any linear function is invertible and a linear function) and (c) the *transitive property*, i.e., $m_1 \sim m_2$ and $m_2 \sim m_3 \leftrightarrow m_1 \sim m_3$ for any metrics $m_1, m_2, m_3$, i.e., the composition of linear functions is still a linear function. We define several properties, i.e., (1) *identity of indiscernibles* (IoI); (2) *symmetry* (or *independence from a ground truth*); (3) *robustness* (Type-I and Type-II); (4) *stability* with respect to $k$; (5) *sensitivity* and *width-swap-dependency*; (6) (induced) *distance*. Most ranking evaluation metrics properties are conserved under linear equivalence (see Section 7). We underline that some of these properties have been defined in diverse contexts, e.g., Gösgens et al. (2021b;a); Hassanzadeh & Milenkovic (2014); Cook et al. (1986); Fligner & Verducci (1986), often under different names. Particularly, Hassanzadeh & Milenkovic (2014) state that the only metric that can be considered a metric on rankings is Kendall's $\tau$. We will show

---

[1] https://anonymous.4open.science/r/rankingsmetricsproperties/README.md

that this is non-compatible with our definition of *identity of indiscernibles*, thus not allowing us to consider it a distance.

## 6.1  Identity of indiscernibles

Given two distinct permutations $\sigma, \tau \in S_n$, a ranking evaluation metric $m$ evaluates how *close* they are. We can easily incur in situations where $\sigma$ and $\tau$ are *so close* to each other to be evaluated as identical by $m$. In contexts like fair ranking aggregation, it is fundamental to distinguish whether elements of specific categories obtain privileged positions, while in huge dimensional spaces this might be not the case, e.g., feature selection of $k$ most important features. We analyze how effectively a metric $m$ distinguishes two different rankings. A metric that satisfies the injective property reflects the difference among rankings in the scores it assigns to them. We name this property the *identity of indiscernibles* property.

**Definition 4.** *A metric $m$ satisfies the* identity of indiscernible (IoI) property *if, $\forall \sigma \in S_n$ fixed, it holds*

$$m(\sigma, \tau) = m(\sigma, \nu) \Leftrightarrow \tau = \nu, \qquad \forall \tau, \nu \in S_n. \tag{2}$$

Up to renaming the elements, we can rewrite equation 2 as $m(\mathrm{id}, \tau) = m(\mathrm{id}, \nu) \Leftrightarrow \nu = \tau$ where id is the usual identity of $S_n$. For (almost) all ranking evaluation metrics, it is possible to find examples in $S_n$ (even with small $n$) that do not satisfy the IoI property. All set based metrics and metrics @$k$ do not satisfy this property as they consider only the set of retrieved (and relevant) items and not the ordering in which they appear in the ranking. For all confusion matrix based metrics, after fixing a permutation $\sigma = (i\ j) \in S_n$ with $i, j < k$ where $k$ is the number of relevant elements, we easily conclude that $m(\mathrm{id}, \mathrm{id}) = m(\mathrm{id}, \sigma)$; All permutations that can be written as a disjoint composition of cycles $\sigma = \nu_{\text{before } k} \circ \nu_{\text{after } k}$ are examples of permutations where the IoI property fails. Table 3 includes examples where the IoI is not satisfied for the various metrics; the confusion matrix based metrics are grouped in a single column as they behave equivalently. All metrics considered but two do not satisfy the IoI property:

**Proposition 6.1.** *DCG and nDCG are the only two ranking evaluation metrics among the ones considered in this paper satisfying the identity of indiscernibles property.*

*Proof of Proposition 6.1.* as DCG and nDCG differ only for a constant multiplicative factor, we prove the claim only for DCG; For the definitions of DCG and nDCG, we refer to Appendix C.3. Given $\sigma \in S_n$, $\mathrm{DCG}(\sigma) = \sum_{i=1}^{n} \frac{\sigma(i)}{\log_2(i+1)}$. The goal is proving that for any $\sigma_1, \sigma_2 \in S_n$, $\mathrm{DCG}(\sigma_1) = \mathrm{DCG}(\sigma_2) \Leftrightarrow \sigma_1 = \sigma_2$. Without loss of generality, we prove: $\mathrm{DCG}(\mathrm{id}) = \mathrm{DCG}(\sigma) \Leftrightarrow \sigma = \mathrm{id}$ for any $\sigma \in S_n$:

$$\sum_{i=1}^{n} \frac{i}{\log_2(i+1)} = \sum_{i=1}^{n} \frac{\sigma(i)}{\log_2(i+1)} \Leftrightarrow \sum_{i=1}^{n} \frac{i - \sigma(i)}{\log_2(i+1)} = 0. \tag{3}$$

However, proving equation 3 is non straight forward; we prove instead the following

$$\sum_{i=1}^{n} \frac{i - \sigma(i)}{\log_2(i+1)} < 0 \Leftrightarrow \sigma \neq \mathrm{id} \in S_n. \tag{4}$$

The equation 4 is a stronger statement than equation 3. We base our proof on induction over the $\mathcal{N}$ size.

**Base case:** The base case $n = 2$ is trivial as $S_2 = \{\mathrm{id}, \sigma = (1\ 2)\}$; in particular, $\mathrm{DCG}(\mathrm{id}) = 0$ while $\mathrm{DCG}(\sigma) = \frac{1 - \sigma(1)}{\log_2 2} + \frac{2 - \sigma(2)}{\log_2 3} = -\frac{1}{\log_2 2} + \frac{1}{\log_2 3} < 0$.

**Inductive case:** The claim holds for $n-1$ and we prove it for $n$; consider $\sigma \in S_n$. We distinguish two cases.

$\sigma$ **fixes one element:** Up to renaming the elements, we suppose that $n$ is fixed by $\sigma$, i.e., $\sigma(n) = n$. Given $n, k \in \mathbb{N}$, we can construct an immersion $i_{n,k} : \sigma \in S_n \mapsto i_{n,k}(\sigma) \in S_{n+k}$ of $S_n$ in $S_{n+k}$, such that $i_{n,k}(\sigma) = \sigma(i)$ if $i \leq n$ otherwise $i_{n,k}(\sigma) = i$; $i_{n,k}$ is injective and surjective on $A = \{\sigma \in S_{n+k} \mid \sigma(i) = i, \forall i > n + k\}$ and $\sigma$ fixes $n$, $\sigma$ belongs to $S_{n-1}$ (as the counter-image of $i_{n,1}$). Therefore, the claim holds.

| ranking length | relevant | baseline | $\sigma$ | $\tau$ | CMB metrics | MSE | RMSE | MAE | MAPE | SMAPE | $R^2$ score | Kendall's $\tau$ | Spearmann $\rho$ | DCG | nDCG | MRR | GMR | NDPM | meanRank |
|---|---|---|---|---|---|---|---|---|---|---|---|---|---|---|---|---|---|---|---|
| 10 | 5 | id | (1 2) | id | ○ | ● | ● | ● | ● | ● | ● | ● | ● | ● | ● | ● | ● | ● | ● |
| 10 | 5 | id | (1 2) | (3 4) | ○ | ○ | ○ | ○ | ● | ● | ○ | ○ | ○ | ● | ● | ○ | ○ | ○ | ○ |
| 10 | 5 | id | (1 2) | (2 4) | ○ | ● | ● | ● | ○ | ○ | ● | ● | ● | ● | ● | ○ | ○ | ● | ○ |

Table 3: Examples of rankings that metrics cannot distinguish. We compare for each evaluation metric $m$ the values $m(\text{id}, \sigma)$ and $m(\text{id}, \tau)$. If the metric fails in distinguishing the two rankings, we impute a ○; else, a ●.

**$\sigma$ does not fix any element:** It holds $\sigma(n) \neq n$ and we can rewrite $\sigma$ as the composition of two permutations, i.e., $\sigma = \tau \circ \mu$ such that $\tau = (j\ n)$ for some fixed $j$ and $\mu$ such that $\mu(s) = \sigma(s)$ if $s \neq n, k^*$, $\mu(s) = j$ if $s = k^*$ and $\mu(s) = n$ if $s = n$ where we named $k^* = \mu^{-1}(j) = \sigma^{-1}(n)$. We can now rewrite $\sigma$ in terms of $\tau \circ \mu$;

$$\sum_{i=1}^{n} \frac{i - \sigma(i)}{\log_2(i+1)} = \sum_{i=1, i \neq k^*}^{n-1} \frac{i - \sigma(i)}{\log_2(i+1)} + \frac{k^* - \sigma(k^*)}{\log_2(k^*+1)} + \frac{n - \sigma(n)}{\log_2(n+1)} =$$

$$\sum_{i=1, i \neq k^*}^{n-1} \frac{i - \mu(i)}{\log_2(i+1)} + \frac{k^* - \tau \circ \mu(k^*)}{\log_2(k^*+1)} + \frac{n - \sigma(n)}{\log_2(n+1)} + \frac{k^* - \mu(k^*)}{\log_2(k^*+1)} - \frac{k^* - \mu(k^*)}{\log_2(k^*+1)} =$$

$$\sum_{i=1}^{n-1} \frac{i - \mu(i)}{\log_2(i+1)} + \frac{k^* - \tau(j)}{\log_2(k^*+1)} + \frac{n - \sigma(n)}{\log_2(n+1)} - \frac{k^* - \mu(k^*)}{\log_2(k^*+1)} =$$

$\sum_{i=1}^{n-1} \frac{i - \mu(i)}{\log_2(i+1)}$ is negative for the inductive hypothesis and momentarily assumes that $\mu \neq \text{id} \in S_{n-1}$, By substituting $\sigma = \tau \circ \mu$, we conclude the proof if we can upper bound their sum with 0.

$$\frac{k^* - \tau(j)}{\log_2(k^*+1)} + \frac{n - \sigma(n)}{\log_2(n+1)} - \frac{k^* - \mu(k^*)}{\log_2(k^*+1)} = \frac{k^* - n - (k^* - \mu(k^*))}{\log_2(k^*+1)} + \frac{n - \sigma(n)}{\log_2(n+1)} =$$

$$\frac{\mu(k^*) - n}{\log_2(k^*+1)} + \frac{n - \sigma(n)}{\log_2(n+1)} < \frac{\mu(k^*) - n}{\log_2(k^*+1)} + \frac{n - \sigma(n)}{\log_2(k^*+1)} = \frac{\mu(k^*) - n + n - \sigma(n)}{\log_2(k^*+1)} = \frac{j - j}{\log_2(k^*+1)} = 0$$

where we used $log_2(n+1) > log_2(k^*+1)$, $\sigma(n) = \tau \circ \mu(n) = \tau(n) = j$ and $\mu(k^*) = j$. Thus, the claim is proved for $\mu \neq \text{id}$. In the case $\mu = \text{id}$: Then it holds $\sigma = \tau$ and $\text{DCG}(\sigma)$ reads

$$\text{DCG}(\sigma) = \sum_{i=1}^{n} \frac{i - \sigma(i)}{\log_2(i+1)} = \sum_{i=1}^{n} \frac{i - \tau(i)}{\log_2(i+1)} =$$

$$\frac{j - \tau(j)}{\log_2(j+1)} + \frac{n - \tau(n)}{\log_2(n+1)} = \frac{j - n}{\log_2(j+1)} + \frac{n - j}{\log_2(n+1)} < \frac{j - n + (n - j)}{\log_2(j+1)} = 0$$

Table 3 clearly shows examples where the IoI property is not satisfied for all the other ranking evaluation metrics; Thus, we conclude that DCG and nDCG are the only two ranking evaluation metrics satisfying the IoI. This concludes the proof. $\square$

Axioms defining distances among partial ordering of items have been defined in Cook et al. (1986); given the matrix representation of partial orderings defined, the authors prove the existence of a unique distance for the specific context.

## 6.2 Symmetry property

In some cases, e.g., RS and IR, the aim is to obtain a ranking as close as possible to a ground truth order. In other applications, e.g., (fair) rank aggregation, the objective is to get a score that reflects how similar the

two rankings are; hence, it is interesting to investigate whether the scores are independent of which of the two is the ground truth. This second case embeds the first one, although it is more generic and fits well with the metrics definition on symmetric groups.

**Definition 5.** *A ranking evaluation metric $m : S_n \times S_n \to \mathbb{R}$ is* symmetric *if $m(\sigma, \nu) = m(\nu, \sigma), \forall \sigma, \nu \in S_n$.*

Although symmetry looks trivial, many ranking evaluation metrics do not satisfy it. All metrics relying on a ground truth ranking can not satisfy the symmetry property; Swapping the two rankings is meaningless in this context as it is equivalent to changing the ground truth.

**Confusion matrix based metrics:** These metrics rely on ground truth labels. However, when the number of relevant coincides with the number of retrieved items, the confusion matrix is symmetric, and the metrics also satisfy the *symmetry property*.

**Correlation based metrics:** Directly from their definition, all the correlation measures are symmetric.

**Cumulative gain based metrics:** They take one ranking and return a corresponding score implicitly comparing with an underlying ordering, which ranks first elements with higher relevance. Thus, they are automatically excluded from being symmetric.

**Error based metrics:** From their definitions, it follows that it is not important which among the two is the ground truth permutation, and the two rankings are interchangeable.

For correlation based and error based metrics, it is easy to prove that they satisfy the symmetry property by substituting in their definition the two orderings of interest $\sigma$ and $\nu$; from their definitions, proving that $m(\sigma, \nu) = m(\nu, \sigma)$ is trivial.

In conclusion, all metrics involving a ground truth are not symmetric; Comparing with a ground truth ranking is often essential in some applications, while when looking for a fair comparison among rankings, it is often preferable to use symmetric evaluation metrics instead of relying on ground truths. The symmetry property is essential and also studied in other contexts, for example in Gösgens et al. (2021a) and Gösgens et al. (2021b).

### 6.3 Robustness

Given two permutations $\tau, \nu \in S_n$, a ranking evaluation metric $m$ reflects how similar $\nu$ and $\tau$ are. We refer with *robustness properties* to a series of properties evaluating the resistance of a ranking evaluation metric to small changes in the rankings. We expect, in principle, that if $\tau$ and $\sigma$ differ only by a swap, they are not evaluated as far from each other as in the case that they differ by an entire cycle containing several elements.

**Definition 6.** *We say that a ranking evaluation metric is* Type I Robust *if a* small change *in one of the rankings implies small changes in its evaluation.*

Given two rankings $\sigma, \nu$ in $S_n$ and $i, j \leq n$, we will consider two types of small changes in rankings:

**Single swaps.** We evaluate how the swap of two elements $i, j$ in the ranking is impacting the evaluation metric, i.e., the absolute value of the difference among the two results

$$|m(\sigma, \nu) - m(\sigma, \nu \circ (i \; j)|; \tag{5}$$

**Sliding of the ranking.** We evaluate how a sliding, i.e., a cycle of the $n$ elements $FC_n = (1 \; 2 \; \cdots \; n)$, impacts the evaluation metric. We evaluate then the difference in absolute value

$$|m(\sigma, \nu) - m(\sigma, \nu \circ FC_n)|. \tag{6}$$

In Table 2, we report the results for the Type I Robustness on 1000 different randomly drawn pair of rankings with lengths 10, 50, and 100. We average the absolute value from equation 5 and equation 6 over the trials and report the approximated results. When we observe only minimal differences from zero, we use $>$ and $>>$ to indicate their approximated entity and round the numbers using two decimals.

**Definition 7.** *We say that a ranking evaluation metric is* Type II Robust *if it is an invariant concerning the composition of permutations, i.e., it holds* $m(\mu, \sigma) = m(\mu \circ \nu, \sigma \circ \nu), \forall \sigma, \nu \in S_n$.

This property was mentioned in Diaconis (1988) as *right-invariance* and in Hassanzadeh & Milenkovic (2014) (Axiom I.2) as resistance to item relabelings; in Hassanzadeh & Milenkovic (2014) also the *left-invariance* is considered as potentially useful. Using the cycle decomposition theorem, we limit to the case $\nu = (j\ k) \in S_n$; *Type II Robustness* property investigates whether a change in the importance ordering in both rankings eventually affects their evaluation. We expect this to be the case when the ranking position is considered a relevance score, particularly in the case of cumulative gain metrics.

**Proposition 6.2.** MSE, RMSE, MAE, MAPE, $R^2$ *score, Kendall's* $\tau$ *score and* Spearmann's $\rho$ *are the only metrics considered in this paper satisfying* $m(\sigma, \nu) = m(\sigma \circ (j\ k), \nu \circ (j\ k)), \forall \sigma, \nu \in S_n$.

*Proof of Proposition 6.2.* **MSE, RMSE, MAE, MAPE, $R^2$ score:** decomposing the sum in the definition of $MSE(\sigma \circ (j\ k), \nu \circ (j\ k))$ among addends involving $k$ or $j$ and others, it is easy to get to $MSE(\sigma, \nu)$. Similarly, for the other metrics. **Kendall's $\tau$:** it is enough to note that the number of discordant and concordant pairs does not change when applying a swap to both the rankings $\sigma$ and $\nu$. **Spearmann's $\rho$:** similarly to the case of the error based metric, we decompose the sum defining the Spearmann's $\rho$ in elements involving $j$ and $k$ and others; manipulating the definition, we eventually get the thesis. **Unicity:** For all the other metrics, finding pairs of rankings providing counterexamples is trivial. For cumulative gain based metrics, the swaps change the association between the position in the ranking and the relevance score. For confusion matrix based metrics, swaps change both the set of relevant and retrieved elements (but not equally); Thus, the evaluation is different after applying swaps in both rankings. □

### 6.4 Sensitivity

The sensitivity property is particularly useful in application to high dimensional spaces where rankings are not fully explored, e.g., in RS and IR methods. RS methods suggest elements in high dimensional space to the users in order of importance, where the first ranked elements correspond to the first suggestions. The users often do not explore the rankings fully; Hence, the relevant information must be available among the first-ranked elements. Many evaluation metrics measure the ability of the RS to return a partially correct ranking of the first $k$ items relying on the fact that the sensitivity of the metric to the permutations @$k$ is intuitively more meaningful than a precise comparison among the complete rankings. We briefly summarize the behavior of the various cluster of metrics.

**confusion matrix based metrics** all are metrics@$k$ and set based metrics; Thus, the ordering of elements before and after $k$ does not matter.

**cumulative gain based metrics** are explicitly based on the position in the rankings; Hence, they are sensitive to positional changes.

**correlation based and error based metrics** being all classified as flat metrics, they equally evaluate the ordering before and after an arbitrary index $k$.

We introduce the definition of *width swap dependency*, formalizing a property that prevents the metrics from being sensitive to positions in the rankings.

**Definition 8.** *Given a swap* $(i\ j) \in S_n$ *and* $|i - j|$ *its* width, *$m$ is* width swap dependent *(WSD) if it evaluates equally swaps with the same width; otherwise, it is called* non-width swap dependent.

**Lemma 6.3.** *The correlation based metrics are* width swap dependent.

*Proof of Lemma 6.3.* **Spearman's rank correlation coefficient** has an equivalent formulation dependent only on the differences $d_i = \sigma(i) - \nu(i)$; The fact that the elements appearing in the ranking are all distinct implies the WSD property directly. To prove the claim for **Kendall's $\tau$** (NDPM is similar), we an arbitrary $n$ and a swap $(i\ j) \in S_n$ of width $d$. We proceed by induction on $d$ and prove that Kendall's $\tau$ is based only

on $d$ independently from $i$ and $j$. If $d = 1$, then the swap is of the form $(i \ i+1)$; in this case, the number of concordant pairs is $\binom{n}{2} - 1$, and the only discordant pair is given by $(i \ i+1)$. Recalling the definition of Kendall's $\tau$, we want to prove that $K_\tau = \frac{|\{\text{concordant pairs}\}| - |\{\text{discordant pairs}\}|}{\binom{n}{2};} = \frac{\binom{n}{2} - 4|i-j| + 2}{\binom{n}{2}}$. This holds for $d = 1$ as $K_\tau(id, (i \ j)) = \frac{\binom{n}{2} - 1 + (\binom{n}{2} - (\binom{n}{2} - 1))}{\binom{n}{2}} = \frac{\binom{n}{2} - 2}{\binom{n}{2}}$. We now suppose that it holds for $d$ and prove it for $d + 1$; the number of discordant pairs in a swap of length $d + 1$ equals the number of elements that are not anymore concordant with $i$, i.e., $d + 1$, plus the number of elements that are not anymore concordant with $j$ minus 1, i.e., $d$; summing up we get $K_\tau(id, (i \ j)) = \frac{\binom{n}{2} - (2d+1) + (\binom{n}{2} - (\binom{n}{2} - (2d+1)))}{\binom{n}{2}} = \frac{\binom{n}{2} - 4(d+1) + 2}{\binom{n}{2}}$. We conclude that Kendall's $\tau$ is *width-swap-dependent*. □

**Definition 9.** *Consider $i, j, k, l \in \{1, \ldots, n\}$ such that $i < j < k < l$ and $(i \ j), (l \ k)$ having the same width. A ranking evaluation metric $m$ is* sensitive *if the swap $(i \ j)$ has a different impact on the metric than $(k \ l)$ in the evaluation metric.*

This property evaluates if a metric assigns more importance to the upper part of the ranking, hence, being particularly useful when $n$ is large. For each metric and each pair of disjoint swaps, we determine whether the metrics evaluate differently swaps happening at various stages in the ranking; The results are summarized in Table 2. The sensitivity property might be necessary for evaluating feature selection and IR/RS techniques, while it is less critical for rank aggregation evaluation.

## 6.5 Stability

Evaluating rankings @$k$ might be tricky; if there is a huge difference between the evaluation @$k$ and @$k+1$, the rankings are not assured to be similar as $k$ could be used as a hyperparameter. As trust and fairness gained importance in the last years, non-stable evaluations must also be tackled. The stability property asks whether a ranking evaluation metric is robust when including additional elements among the relevant items.

**Definition 10.** *A ranking evaluation metric $m$ is* stable *if, for any two rankings $\sigma, \nu \in S_n$, it holds $|m_{@k}(\sigma, \nu) - m_{@k+1}(\sigma, \nu)| < \epsilon_k$ with $\epsilon_k$ small. Moreover, the sequence $\{\epsilon_k\}_k$ satisfies $\lim_{k \to n} \epsilon_k = 0$.*

For large $k$, the differences between the evaluations @$k$ and @$k+1$ wiggle around zero. We evaluate if it is possible to approximate $\epsilon_k$ with $\frac{1}{k}$ for each $n \in \mathbb{N}$; In Table 2, we report the results of the conducted experiments. We randomly draw 1000 pairs of rankings in $S_{1000}$; for each pair, we compute the absolute values as stated in 10 and average the results over the number of trials; we finally count the number of times that 10 holds with $\epsilon_k = \frac{1}{k}$. As a criterion for a metric to be stable, we used that it should be satisfied in 97.5% of the cases. For metrics where the number of relevant elements is not essential, including the error based metrics, we got that 10 is satisfied in all the cases.

## 6.6 Distance

In mathematics, the terms metric and distance are considered synonymous. This section discusses whether the ranking evaluation metrics define a distance notion on symmetric groups. Also Diaconis (1988) and Hassanzadeh & Milenkovic (2014) mentioned the importance of having distances on symmetric groups. We show that most of them are not metrics in the mathematical sense and further investigate whether they induce distances.

**Definition 11.** *A* distance *(or mathematical* metric*) on a set $X$ is a function $d : X \times X \to [0, \infty) : (x, y) \mapsto d(x, y) \in \mathbb{R}_+$ such that for all $x, y, z \in X$, (1) the identity of indiscernibles, i.e., $d(x, y) = 0 \Leftrightarrow x = y$, (2) the symmetry, i.e., $d(x, y) = d(y, x)$, and (3) the triangle inequality, i.e., $d(x, y) \leq d(x, z) + d(z, y)$ are satisfied.*

**Definition 12.** *A ranking evaluation metric $m$ on $S_n$ is* linearly transformable *into a distance if there exists a linear function $f$ such that $f_m(\sigma, \nu) = f(m(\sigma, \nu)) \forall \sigma, \nu \in S_n$ and $f_m$ is a distance.*

We know a priori that any evaluation metric not satisfying the *identity of indiscernibles* or the *symmetry* properties is not a distance; furthermore, we show in Section 7 that it is not even linear equivalent to a distance. We limit our study to ranking evaluation metrics for which the first two properties hold and check

whether the triangle inequality is also satisfied. We distinguish two cases based on the sign of the coefficient defining the linear transformation and, in some cases, limit to *positively linear equivalent* metrics where all coefficients are positive real numbers.

**Definition 13.** *A function $f_m$ is* positive definite *if it holds $f_m(\sigma, \nu) \geq 0, \forall \sigma, \nu \in S_n$.*

Given Definition 3, to check whether $f_m$ is positive definite it is enough to check whether $f_m$ is a bounded function; given that $m$ satisfies the maximal agreement property, its linear equivalent $\tilde{m}$ defined as

$$\tilde{m}(\sigma, \nu) = m_{\max} - m(\sigma, \nu) \tag{7}$$

satisfies the positive definiteness property. A change in the ordering in which rankings are evaluated by $\tilde{m}$ is a consequence of the change of sign in equation 7; this implies that the number of disagreements among metrics is reversed but still preserves the consistency definition. We refer to metrics satisfying both the maximal and minimal agreement properties as *bounded*. Ideally, $m$ satisfies $m(\sigma, \nu) = m_{\max}$ if $\sigma = \nu$ and $m(\sigma, \nu) = m_{\min}$ if $\nu$ is the reversed order of $\sigma$ (see Appendix A). We consider two dimensional functions $f : S_n \times S_n \to \mathbb{R}$, where $f$ either refers to a ranking evaluation metric $m : S_n \times S_n \to \mathbb{R}$ or the induced function $f_m : S_n \times S_n \to \mathbb{R}$ in the case that $m : S_n \to \mathbb{R}$.

**Definition 14.** *A function $f_m$ satisfies the* triangle inequality *if, $\forall n \in \mathbb{N}$, it holds $f_m(\sigma, \mu) \leq f_m(\sigma, \nu) + f_m(\nu, \mu), \forall \sigma, \nu, \mu \in S_n$.*

Given $m$, we consider two options as potential induced distances, i.e., $f_m(\sigma, \nu) = m(\sigma) - m(\nu)$ or $\tilde{f}_m(\sigma, \nu) = |m(\sigma) - m(\nu)|$. DCG and nDCG are the only two metrics satisfying the IoI property essential for a metric being a distance. Hence, we limit our study to DCG and nDCG; Moreover, they are linearly equivalent and it is sufficient to prove the result only for one of them.

**Proposition 6.4.** *$f_m$ is* not *a distance while $\tilde{f}_m$ is a distance, where $m$ is either DCG or nDCG.*

*Proof of Proposition 6.4.* We must prove the three properties defining a distance for $m = $ DCG.

**Identity of Indiscernibles property:** Proposition 6.1 states that DCG satisfies the IoI property. It follows that $f_m(\sigma, \nu) = 0 \Leftrightarrow \sigma = \nu$; Similarly, $\tilde{f}_m(\sigma, \nu) = 0 \Leftrightarrow \nu = \sigma$.

**Symmetry property:** It is easy to find pairs of permutations $\sigma, \nu \in S_n$ such that $f_{DCG}(\nu, \sigma) = f_{DCG}(\sigma, \nu)$; In particular, $f_{DCG}$ satisfy the anti-symmetric property, i.e., $f_{DCG}(\nu, \sigma) = DCG(\nu) - DCG(\sigma) = -[DCG(\sigma) - DCG(\nu)] = -f_{DCG}(\sigma, \nu)$. On the other hand, $\tilde{f}_{DCG}$ satisfies the symmetry property.

**Triangle inequality:** The triangle inequality property is satisfied if $\forall \nu, \sigma, \mu \in S_n$ holds $f_{DCG}(\sigma, \mu) \leq f_{DCG}(\sigma, \nu) + f_{DCG}(\nu, \mu)$. Expanding the formula of DCG we get

$$f_{DCG}(\mu, \sigma) = DCG(\mu) - DCG(\sigma) = DCG(\mu) - DCG(\nu) + DCG(\nu) - DCG(\sigma) = f_{DCG}(\mu, \nu) + f_{DCG}(\nu, \sigma);$$

The equality holds $\forall \nu, \sigma, \mu \in S_n$; for $\tilde{f}_{DCG}$, the property still holds with the inequality:

$$\tilde{f}_{DCG}(\mu, \sigma) = |DCG(\mu) - DCG(\sigma)| = |DCG(\mu) - DCG(\nu) + DCG(\nu) - DCG(\sigma)| \leq$$
$$\leq |DCG(\mu) - DCG(\nu)| + |DCG(\nu) - DCG(\sigma)| = \tilde{f}_{DCG}(\mu, \nu) + \tilde{f}_{DCG}(\nu, \sigma).$$

**Positive definiteness:** $\tilde{f}_{DCG}$ is defined as an absolute value; the claim obviously holds. Instead, $f_{DCG}$ can assume both positive and negative values. This concludes the proof. □

## 7 Relation among the properties

We introduced some desirable properties for ranking evaluation metrics: each of them considering different aspects of the metrics, we can prove that they interact at certain levels. In particular, the *distance property* is a summary of three different properties: the *symmetry* and *maximal agreement property*, introduced in the previous sections, and the *triangle inequality* property. Therefore, whenever one of the first two properties is not satisfied, it is meaningless to check whether the triangle property holds. Furthermore, the *maximal agreement property* (Definition 15) is equivalent to the *positive definiteness*, i.e., the values assigned by $m$ are all non-negative. Most properties are satisfied by the metrics up to linear equivalence.

**Proposition 7.1.** *Given $\tilde{m}$ and $m$ ranking evaluation metrics, if $\tilde{m}$ and $m$ are linearly equivalent then (1) if $m$ satisfies the maximal agreement property, also $|\tilde{m}|$ does; if $m$ is symmetric, $\tilde{m}$ is also symmetric; if $m$ satisfies the IoI property, also $\tilde{m}$ does; (2) $\tilde{m}$ and $m$ are consistent.*

*Proof of Proposition 7.1.* (1) We know that $m$ and $\tilde{m}$ are linear equivalent, then it exists a function $f$ and $a, b \in \mathbb{R}$ such that $\forall \sigma, \nu \in S_n$, $\tilde{m}(\sigma, \nu) = f(m(\sigma, \nu)) = a \cdot m(\sigma, \nu) + b$.

**Maximal agreement** We need to distinguish two cases: $a > 0$ and $a < 0$. The case $a = 0$ is trivial. If $a > 0$, for any rankings $\sigma$, $\nu$, it holds $\tilde{m}(\sigma, \sigma) = a \cdot m(\sigma, \sigma) + b \geq a \cdot m(\sigma, \nu) + b = \tilde{m}(\sigma, \nu)$. Thus, $\tilde{m}$ satisfies the maximal agreement property. Similarly, if $a < 0$, $\tilde{m}(\sigma, \sigma) = a \cdot m(\sigma, \sigma) + b \leq a \cdot m(\sigma, \nu) + b = \tilde{m}(\sigma, \nu)$. Thus, $\tilde{m}(\sigma, \sigma) \leq \tilde{m}(\sigma, \nu)$ holds $\forall \sigma$, $\nu$, i.e., $-\tilde{m}$ satisfies the maximal agreement property.

**Symmetry** If $m$ is symmetric this means that $\forall \sigma$, $\nu$ orderings, $m(\sigma, \nu) = m(\nu, \sigma)$. Then $\tilde{m}(\sigma, \nu) = a \cdot m(\sigma, \nu) + b = a \cdot m(\nu, \sigma) + b = \tilde{m}(\nu, \sigma)$ proving that $\tilde{m}$ is symmetric too.

**Identity of indiscernibles** If $m$ satisfies $m(\sigma, \nu) = m_{\max} \leftrightarrow \sigma = \nu$, substituting the condition in the linear equivalence, we get that $\tilde{m}$ satisfies the IoI property too; the existence of $m_{\max}$ is unnecessary.

(2) it is an obvious consequence of the definition of monotone functions. For any $x_1, x_2$ in $\mathrm{dom}(f)$, $f$ is a monotone increasing function if $x_1 \geq x_2$ implies $f(x_1) \geq f(x_2)$ in the case of positive linear equivalence. The same holds with $\leq$ for decreasing monotone function. Thus this concludes the proof. $\square$

**Proposition 7.2.** *The identity of indiscernibles implies the maximal agreement property.*

*Proof.* To prove that the opposite is not valid, it is enough to find a ranking evaluation metric that satisfies the maximal agreement property. Still, the IoI does not hold. The precision gives a trivial example; although the maximal agreement property is satisfied (or equivalently, as precision@$k$ is positive definite), we saw various examples where the IoI is not satisfied. On the other hand, consider a ranking evaluation metric $m$ and and its linear equivalent metric $\tilde{m}(\sigma, \mu) = m_{\max} - m(\sigma, \mu)$ for any $\sigma, \mu \in S_n$; $\tilde{m}$ satisfies the IoI property if $\tilde{m}(\sigma, \mu) = 0 \leftrightarrow \sigma = \mu$. $\square$

# 8 Conclusion and discussion

We provide theoretical and experimental insights on the necessity of careful choices for ranking evaluation on symmetric groups; We showed that non-consistent evaluations appear when using ranking evaluation metrics and proposed theoretical properties allowing for a deeper understanding of these metrics. We illustrated how most metrics do not distinguish small changes among rankings, how single swaps and slides of the rankings influence their evaluation, and how robust the metrics are. We additionally gave insights on the implications among the defined properties and tried to obtain a distance on the symmetric groups. We defined several mathematical properties, each highly desirable in some contexts and less in others. The IoI property is desirable when looking for metrics highly sensitive to small changes, such as fairness in rankings or top $k$ items in RS. Conversely, Robustness assures that small changes do not have a huge impact on the evaluations. Combining these two properties assures contemporaneously that small changes are not overlooked but do not significantly impact the scores. When a ground truth ranking is not available, it is important to use only symmetric metrics deriving from ground truth the symmetry. Sensitivity is crucial for RS and IR techniques evaluation, where changes in the top part of the rankings are more influential than in the lower part. Stability is generally important in evaluating rankings @$k$; the proposed analysis gives a general idea of the ability of metrics to generate a stable evaluation. However, we recommend considering evaluating the impact @$k$ and @$(k + i)$ with $i$ arbitrarily chosen, in particular, when $k << n$. Finally, the distance property is defined to complete the proposed analysis and highlights the chance that mathematical terms are misused in many machine learning contexts.

Despite the rough evaluation of some metrics, they are used in the literature as one of the most powerful techniques to evaluate RS, IR, feature selection, and rank aggregation methods; examples are the confusion matrix-based metrics that do not allow for precise comparisons among orderings. On the other hand, metrics based on errors satisfy most of the proposed properties but are rarely used for rankings. Cumulative

gain-based metrics offer a good compromise among correlation and confusion matrix based metrics; the necessity of ground truth and relevance labels, however, is their biggest weakness. Having collected the obtained theoretical and experimental results in a concise table, we allow for insights of immediate use.

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

# A    Are the metrics interpretable?

Given the importance of trust, fairness, and explainability for machine learning methods, one could then ask how *interpretable* the scores assigned by the metrics are. We first need some definitions.

**Definition 15.** *A ranking evaluation metric $m$ is said to satisfy the* maximal agreement property *if (a) $m(\sigma, \sigma) = m_{\max}, \forall \sigma \in S_n$ and (b) $m(\sigma, \nu) \leq m_{\max}, \forall \nu, \sigma \in S_n$. We say that $m$ is* lower-bounded *if it exists a real number $m_{\min}$ such that $m(\sigma, \nu) \geq m_{\min}, \forall \nu, \sigma \in S_n$. An evaluation metric that admits a lower bound is said to satisfy the* minimal agreement *property.*

We define some properties for metrics to be interpretable, i.e., (1) each ranking is maximally similar to itself and, given $n \in \mathbb{N}$, this value is constant (we refer to it with $m_{\max}$), i.e., $m(\sigma, \sigma) = m_{\max}, \forall \sigma \in S_n$; (2) $m$ satisfies the maximal agreement property; (3) there exists a lower bound $m_{\min}$ for any possible pair of rankings, i.e., $m(\sigma, \mu) \geq m_{\min}, \forall \sigma, \mu \in S_n$.

The maximal agreement property says that each ranking is maximally similar to itself, and no other ranking can achieve a higher score than $m_{\max}$. Property (1) states that $m_{\max}$ is independent of the length of the rankings. Together with the maximal agreement, it implies that a ranking evaluation metric is a monotone increasing function of the similarity of two rankings: the more similar two rankings are, the higher the score they get when evaluated using an 'interpretable' metric. If $m_{\max}$ is independent of the rankings' length, we can compare the similarity among rankings independently of the size of the rankings themselves. However, this property is hardly satisfied by any metrics; each metric can be normalized such that $m_{\max}$ results independent from $n$. The only metrics, among the ones considered in this paper, automatically satisfying this property are Kendall's $\tau$ score and Spearmann $\rho$. We underline that for some metrics, e.g., error-based metrics, the lowest scores are assigned to maximally similar pairs of rankings; it can be tested whether linear transformations of these metrics through equation 7 allow satisfying the aforementioned properties.

A ranking evaluation metric satisfying the maximal agreement property is also *upper-bounded*. For the sake of interpretability, we could check whether a metric $m$ satisfies $m(\rho^{-1}, \rho) = m_{\min}$ where $\rho^{-1}$ indicates the inverse ranking. However, this is not true for most ranking evaluation metrics. Kendall's $\tau$ satisfies this property. However, it is already questionable which is the inverse of a ranking, i.e., if the furthest possible ranking is the one ordering first the last elements and last the first elements; Using the inverse of the ranking in the symmetric group operation ∘ also does not provide an excellent practical alternative. Assessing whether metrics for permutations are humanly interpretable is not new and has already been discussed in Diaconis (1988); However, then as well as now, the concept of interpretability lacks a unified definition of what interpretability means, a common issue in most cases where interpretability found interest and application. Thus we leave this section open and do not argue further on the interpretability of the criteria defined.

# B    Proofs

*Extended proof of Lemma 6.3.* The **Spearman's rank correlation coefficient** is *width swap dependent*; this is shown directly by the equivalent formulation of the metric dependent only on the differences $d_i = \sigma(i) - \nu(i)$ additionally knowing that the elements appearing in the ranking are all distinct.

We prove the proposition for **Kendall's Tau** (for NDPM it is similar); We fix the symmetric space $S_n$ for an arbitrary natural number $n$ and a swap $(i\ j) \in S_n$ of width $d$. We proceed by induction on the width $d$ and prove that Kendall's $\tau$ is based only on $d$ independently from $i$ and $j$.

If $d = 1$, then the swap is of the form $(i\ i+1)$; recall the definition of Kendall's Tau

$$K_\tau = \frac{|\{\text{concordant pairs}\}| - |\{\text{discordant pairs}\}|}{\binom{n}{2}};$$

hence, in this case, the number of concordant pairs is $\binom{n}{2} - 1$, and the only discordant pair is given by $(i\ i+1)$. We want to prove that

$$K_\tau = \frac{\binom{n}{2} - 4|i - j| + 2}{\binom{n}{2}}.$$

This holds for $d = 1$ as $K_\tau(id, (i\ j)) = \frac{\binom{n}{2}-1+(\binom{n}{2}-(\binom{n}{2}-1))}{\binom{n}{2}} = \frac{\binom{n}{2}-2}{\binom{n}{2}}$. Suppose that it holds for $d$ and we prove it for $d+1$; the number of discordant pairs in a swap of length $d+1$ equals the number of elements that are not anymore concordant with $i$, i.e., $d+1$, plus the number of elements that are not anymore concordant with $j$ minus 1, i.e., $d$; summing up we get

$$K_\tau(id, (i\ j)) = \frac{\binom{n}{2} - (2d+1) + (\binom{n}{2} - (\binom{n}{2} - (2d+1)))}{\binom{n}{2}} =$$
$$\frac{\binom{n}{2} - 4(d+1) + 2}{\binom{n}{2}}$$

Thus Kendall's Tau is *width-swap-dependent*. $\qquad\square$

*Extended proof of Proposition 6.4.* We have to prove that DCG satisfies the three properties defining a distance.

**Identity of Indiscernibles property:** In Section 6.1 we proved that DCG satisfies the Identity of Indiscernibles property. It follows that

$$f_m(\sigma, \nu) = 0 \Leftrightarrow \sigma = \nu; \tag{8}$$

Similarly, $\tilde{f}_m(\sigma, \nu) = 0$ if and only if $\nu = \sigma$

**Symmetry property:** it is easy to find pairs of permutations $\sigma, \nu \in S_n$ such that $f_{DCG}(\nu, \sigma) = f_{DCG}(\sigma, \nu)$; In particular, we can prove that $f_{DCG}$ satisfy the anti-symmetric property, i.e.,

$$f_{DCG}(\nu, \sigma) = DCG(\nu) - DCG(\sigma) =$$
$$- [DCG(\sigma) - DCG(\nu)] = -f_{DCG}(\sigma, \nu).$$

On the other hand, $\tilde{f}_{DCG}$ satisfies the symmetry property.

**Triangle inequality** The triangle inequality property is satisfied if for any $\nu, \sigma, \mu \in S_n$ holds

$$f_{DCG}(\sigma, \mu) \leq f_{DCG}(\sigma, \nu) + f_{DCG}(\nu, \mu). \tag{9}$$

Expanding the formula of DCG we get

$$f_{DCG}(\mu, \sigma) = DCG(\mu) - DCG(\sigma)$$
$$DCG(\mu) - DCG(\nu) + DCG(\nu) - DCG(\sigma) =$$
$$f_{DCG}(\mu, \nu) + f_{DCG}(\nu, \sigma);$$

The equality holds for each $\nu, \sigma, \mu \in S_n$; in the case of $\tilde{f}_{DCG}$ the property still holds with the inequality:

$$\tilde{f}_{DCG}(\mu, \sigma) = |DCG(\mu) - DCG(\sigma)|$$
$$|DCG(\mu) - DCG(\nu) + DCG(\nu) - DCG(\sigma)| \leq$$
$$|DCG(\mu) - DCG(\nu)| + |DCG(\nu) - DCG(\sigma)| =$$
$$\tilde{f}_{DCG}(\mu, \nu) + \tilde{f}_{DCG}(\nu, \sigma).$$

**Positive definiteness:** $\tilde{f}_{DCG}$ is obviously positive definite because of the absolute value; instead, $f_{DCG}$ can assume both positive and negative values.

This concludes the proof. $\qquad\square$

*Extended proof of Proposition 7.1.* We prove the properties in (1) one by one. We know that $m$ and $\tilde{m}$ are linear equivalent, then it exists a function $f$ such that

$$\tilde{m}(\sigma, \nu) = f(m(\sigma, \nu)) = a \cdot m(\sigma, \nu) + b$$

for any $\sigma$, $\nu$ and some $a, b \in \mathbb{R}$.

**Maximal agreement** We need to distinguish two cases: $a > 0$ and $a < 0$. The case $a = 0$ is trivial. If $a > 0$, for any rankings $\sigma$, $\nu$, it holds

$$\tilde{m}(\sigma, \sigma) = a \cdot m(\sigma, \sigma) + b \geq a \cdot m(\sigma, \nu) + b = \tilde{m}(\sigma, \nu).$$

thus $\tilde{m}$ satisfies the maximal agreement property. If $a < 0$,

$$\tilde{m}(\sigma, \sigma) = a \cdot m(\sigma, \sigma) + b \leq a \cdot m(\sigma, \nu) + b = \tilde{m}(\sigma, \nu).$$

Thus, $\tilde{m}(\sigma, \sigma) \leq \tilde{m}(\sigma, \nu)$ holds for any rankings $\sigma$, $\nu$, i.e., $-\tilde{m}$ satisfies the maximal agreement property; this concludes the proof. As already noted before, either we limit the study to positive linear equivalences, or we need to distinguish multiple cases.

**Symmetry** If $m$ is symmetric this means that for all $\sigma$, $\nu$ orderings, $m(\sigma, \nu) = m(\nu, \sigma)$. Then

$$\begin{aligned}\tilde{m}(\sigma, \nu) &= a \cdot m(\sigma, \nu) + b \\ &= a \cdot m(\nu, \sigma) + b = \tilde{m}(\nu, \sigma)\end{aligned}$$

proving that $\tilde{m}$ is symmetric too.

**Identity of indiscernibles** If $m$ satisfies the identity of indiscernibles property, then

$$\begin{aligned}m(\sigma, \nu) &= m_{\max} \leftrightarrow \sigma = \nu \\ a \cdot m(\sigma, \nu) + b &= a \cdot m_{\max} + b \leftrightarrow \sigma = \nu \\ \tilde{m}(\sigma, \nu) &= \tilde{m}_{\max} \leftrightarrow \sigma = \nu\end{aligned}$$

therefore $\tilde{m}$ satisfies the identity of indiscernibles property. We note that assuming the existence of $m_{\max}$ is unnecessary.

This complete the proof of Proposition 6.1.

To prove (2), it is enough to note that this is an obvious consequence of the definition of monotone functions. For any $x_1, x_2$ in dom($f$), $f$ is a monotone increasing function if $x_1 \geq x_2$ implies $f(x_1) \geq f(x_2)$ in the case of positive linear equivalence. The same holds with $\leq$ for decreasing monotone function. Thus this concludes the proof. □

*Proof of Proposition 7.2.* To prove that the opposite is not true, it is enough to find a ranking evaluation metric that satisfies the maximal agreement property, but the identity of indiscernibles does not hold. A trivial example is given by the precision@$k$; although the maximal agreement property is satisfied (or equivalently, as precision@$k$ is positive definite), we saw various examples where the identity of indiscernibles is not satisfied.

On the other hand, consider a ranking evaluation metric $r$ and its linear equivalent metric given by $\tilde{r}(\sigma, \mu) = c_{\max} - r(\sigma, \mu)$ for any $\sigma, \mu \in S_n$; then $\tilde{r}$ satisfies the identity of indiscernibles property if $\tilde{r}(\sigma, \mu) = 0 \leftrightarrow \sigma = \mu$. □

## C  Metrics' definitions

We give some insights on the metrics we mentioned and analyzed throughout the paper; we also properly define here some of the terms and metrics we used.

## C.1 Confusion matrix based metrics

Consider a set $N$ of $n$ elements, a subset $R \subset N$ of the relevant elements and a subset $S \subset N$ of the retrieved elements; the confusion matrix, i.e., a $C = \mathbb{N}^{2 \times 2}$, is defined such that $C_{1,1} = |S \cap R|$, $C_{1,2} = |S \setminus R|$, $C_{2,1} = |R \setminus S|$ and $C_{2,2} = n - |S \cup R|$. Each metric in this group is defined on the sizes of intersections, unions or differences among the sets $R, S$ and $N$. Given two rankings $\sigma, \tau \in S_n$ and two natural numbers $j, k < n$, we can define the set of relevant elements $R$ being the first $j$ths ranked elements by $\sigma$ and the set of retrieved elements being the first $k$ths ranked elements by $\tau$, i.e., $R = \text{set}\left(\sigma_{|j}\right)$ and $S = \text{set}\left(\tau_{|k}\right)$.

**General properties.** Defined only on set based quantities, the ordering of the elements appearing before and after $j$ (or $k$) is irrelevant. Some of these metrics represent a powerful tool for evaluating and comparing rankings. Their strength is well founded on the simplicity and interpretability of the definitions; however, one should consider more sophisticated evaluation metrics when the interest is in the rankings rather than the ability to retrieve the relevant elements.

We shortly define the used metrics and, for sake of readability, we drop the notation $(\sigma, \tau)$; we consider $k = j$ throughout the manuscript. The *precision* represents the fraction of the number of retrieved elements that are relevant, i.e., precision $= \frac{|R \cap S|}{|S|}$. The *recall* represents the fraction of relevant elements successfully retrieved, i.e., recall $= \frac{|R \cap S|}{|R|}$; It is often referred to as *sensitivity*. The *Fallout* represents the proportion of non-relevant elements that are retrieved, i.e., fallout $= \frac{|(N \setminus R) \cap S|}{|N \setminus R|}$. The *F-score* is the harmonic mean of precision and recall where precision and recall can also be not be evenly weighted $F_\beta = \frac{(1 + \beta^2) \cdot (\text{precision} + \text{recall})}{\beta^2 \text{precision} + \text{recall}}$; if $\beta = 1$ then precision and recall are evenly weighted and we refer to it as F1-score. The *accuracy* is defined as $\text{ACC} = \frac{|S \cap R| + n - |S \cup R|}{n}$. The *Jaccard index* is defined as Jaccard $= \frac{|S \cap R|}{|S \cup R|}$. The *Matthews correlation coefficient (MCC)* is defined as $\text{MCC} = \frac{|R \cap S|(n - |S \cup R|) - |S \setminus R||R \setminus S|}{|S||R|(n - |R|)(n - |S|)}$. Given the quantities $\text{TPR} = \frac{|R \cap S|}{|R|}$, $\text{TNR} = \frac{n - |R \cup S|}{n - |R|}$, $\text{FNR} = 1 - \text{TPR}$, $\text{FPR} = 1 - \text{TNR}$, $\text{FDR} = \frac{|S \setminus R|}{|S|}$ and $\text{NPV} = \frac{n - |R \cup S|}{n - |S|}$, we can also define the *informedness*, i.e., informedness $= \text{TPR} + \text{TNR} - 1$, the *markedness*, i.e., markedness $= 1 - \text{FDR} + \text{NPV} - 1$, the *false omission rate FOR*, i.e., $\text{FOR} = 1 - \text{NPV}$, the *prevalence threshold PT*, i.e., $\text{PT} = \frac{\sqrt{TPR \cdot FPR} - FPR}{TPR - FPR}$, the *Fowlkes–Mallows index FM*, i.e., $\text{FM} = \sqrt{(1 - FDR)TPR}$, the *balanced accuracy* BA, i.e., $\text{BA} = \frac{\text{TPR} + \text{TNR}}{2}$, and finally the *Positive likelihood ratio LR+*, i.e., $\text{LR+} = \frac{\text{TPR}}{1 - \text{TNR}}$ the *Negative likelihood ratio LR-*, i.e., $\text{LR-} = \frac{1 - TPR}{TNR}$.

## C.2 Correlation measures

They explicitly rely on the correlation among the two rankings and are often used in statistical applications. In contrast to confusion matrix based metrics, they consider all the length of the rankings. *Kendall's $\tau$ coefficient* and *Spearmann $\rho$* consider the permutation of the elements over arrays of length $n$ Kendall (1938). The *Kendall's $\tau$ coefficient* is based on the definition of concordant and discordant couples (two elements $i, j$ are *concordant* in $\sigma, \tau$ if $\sigma(i) < \sigma(j)$ and $\tau(i) < \tau(j)$ or the same holds with $>$). In particular,

$$\tau = \frac{|\{\text{concordant pairs}\}| - |\{\text{discordant pairs}\}|}{\binom{n}{2}}$$

Kendall's $\tau$ varies in the interval $[-1, +1]$: $\tau = 1$ if $\sigma$ and $\tau$ agree perfectly while $\tau = -1$ if one ordering is the reverse of the other. Furthermore, if $\sigma$ and $\nu$ are independent then $\tau \approx 0$.
The *Spearmann score* is defined as the Pearson correlation coefficient and in the case that the $n$ ranks are distinct integers, it can be computed using the formula

$$r = 1 - \frac{6 \sum_{i=1}^n (\sigma(i) - \tau(i))^2}{n(n^2 - 1)}.$$

As a drawback, correlation based metrics assign the same importance to the first part as to the last part of the rankings; as they equally evaluate exchanges in the first-ranked items and in the ending part of the ranking, they do not properly fit with evaluating orderings. Both Spearmann's $\rho$ and Kendall's $\tau$ directly

penalize swaps of 'further located' elements; Considering two rankings $\sigma, \tau$ that differ for one single swap $(i\ j)$, if $i, j$ are far in the rankings, then they will evaluate their difference as being bigger as if they would be nearer.

Finally, the *Normalized Distance-based Performance Measure* NDPM from Yao (1995): Given $\sigma, \tau$ and the following quantities

$$C_+ = \sum_{i,j} \text{sgn}(\sigma(i) - \sigma(j)) \text{sgn}(\tau(i) - \tau(j))$$

$$C_- = \sum_{i,j} \text{sgn}(\sigma(i) - \sigma(j)) \text{sgn}(\tau(j) - \tau(i))$$

$$C_u = \sum_{i,j} [\text{sgn}(\sigma(i) - \sigma(j))]^2$$

$$C_s = \sum_{i,j} [\text{sgn}(\tau(i) - \tau(j))]^2$$

$$C_{u0} = C_u - C_+ - C_-$$

from their combination, we get $NDPM(\sigma, \tau) = \frac{C_- + \frac{1}{2} C_{u0}}{C_u}$.

### C.3    Cumulative gain based metrics

Constructed with the specific aim of evaluating whether the ordering of relevant elements is respected, they use relevance scores assigned to each element. We assume that the relevance score of an element $i$ is represented by all different relevant scores; in particular, we consider $\text{rel}_i = \sigma(i)$ allowing to fairly compare with other metrics that do not have access to relevance scores of items, but only their position. We initially considered assigning $\text{rel}_i = 1$ to all relevant elements; however, this immediately would imply that DCG and nDCG do not satisfy the IoI property as relevant elements are indistinguishable. The *Discounted cumulative gain DCG* assumes that highly relevant items appearing lower in a search result list should be penalized as the graded relevance value is scaled to be logarithmically proportional to the position of the item; the definition reads $\text{DCG} = \sum_{i=1}^{n} \frac{\sigma(i)}{\log_2(i+1)}$. The *Normalized discounted cumulative gain nDCG* is a normalization of the DCG through the normalization coefficient IDCG, computed by sorting all elements by their relative relevance, and producing the maximum possible DCG. The two metrics nDCG and DCG are linearly equivalent; Thus, as Figure 1 already empirically showed, there are no inconsistencies among them.

Strictly connected to the cumulative gain metrics is the *Mean Reciprocal Rank* MRR that evaluates the position of each relevant element in the ranking and computes the average of the reciprocal positional ranking of the results, i.e., $\text{MRR} = \frac{1}{|R|} \sum_{i=1}^{|R|} \frac{1}{\sigma(i)}$; thus, it is the relevance scores' *harmonic mean*. The *meanRank* is defined as $\text{meanRank}(\sigma) = \frac{1}{|R|} \sum_{i=1}^{|R|} \sigma(i)$ and the *GMR* as the geometric mean of the first $k$ ranked elements of the ranking $\sigma$.

### C.3.1    Error based metrics

Although meant to evaluate continuous and discrete labels, error-based metrics found application in evaluating rankings. They do not consider the ordering of items but compute the difference in each position, sum it all together, and return an average. We will briefly provide the definitions of each metric in this section. Among them, we find the *mean squared error* MSE, defined as $MSE(\sigma, \tau) = \sum_{i=1}^{n} (\sigma(i) - \tau(i))^2$ for $\sigma, \tau \in S_n$, and the *mean absolute error* MAE, defined as $MAE(\sigma, \tau) = \sum_{i=1}^{n} |\sigma(i) - \tau(i)|$. The *rooted mean squared error* RMSE and the *rooted mean absolute error* RMAE are their respective rooted versions, i.e., $RMSE = \sqrt{MSE}$ and $RMAE = \sqrt{MAE}$. The *symmetric mean absolute percentage error* SMAPE is defined as $SMAPE(\sigma, \tau) = \frac{100}{n} \sum_{i=1}^{n} 2 \frac{|\sigma(i) - \tau(i)|}{\sigma(i) + \tau(i)}$ for $\sigma, \tau \in S_n$. Finally, the R2 score is defined as $\text{R2score} = 1 - \frac{\sum_{i=1}^{n} (\sigma(i) - \tau(i))^2}{\sum_{i=1}^{n} (\sigma(i) - \frac{1}{n} \sum_{j=1}^{n} \tau(j))^2}$.

