# OpenReview forum: "Ranking evaluation metrics from a group-theoretic perspective"
_TMLR — Rejected by TMLR_

### Review · Reviewer_vkRs · 2023-07-20

**Summary Of Contributions:**

This paper tackles the challenge of analyzing ranking evaluation metrics, noting that the choice of metric can often mask weaknesses in the methods being evaluated. The authors propose a novel theoretical approach, grounded in the mathematical formality of symmetric groups, that allows for a more context-independent assessment of these metrics. In particular, the paper introduces an "agreement ratio" to gauge how often pairs of metrics disagree and defines key mathematical properties for these metrics. This theoretical foundation is intended to help users make more informed choices when selecting evaluation metrics for their specific needs and situations. It can potentially bring about a more insightful understanding of the goals and utilization of ranking evaluation metrics, even amidst the challenges of their experimental validation.

**Audience:**

Yes

**Broader Impact Concerns:**

I do not have any broader impact concerns.

**Claims And Evidence:**

Yes

**Requested Changes:**

The paper could be strengthened by including empirical studies, real-world examples, or demonstrations to validate the theoretical findings and show how the proposed approach can be useful to improve ranking system effectiveness.

**Strengths And Weaknesses:**

**Strengths**
- The paper presents a novel theoretical approach based on symmetric groups for analyzing and clustering ranking evaluation metrics, providing a more context-independent way to assess and interpret these metrics.


- The paper introduces an "agreement ratio" as a measure of how often pairs of metrics disagree, which could be a useful tool in understanding the inconsistencies in evaluation results.


- The paper and provided framework potentially allow for a more informed and conscious choice of evaluation metrics, which could improve the assessment and comparison of machine learning techniques.

**Weaknesses**
- The paper heavily relies on mathematical and theoretical concepts, which might make it difficult for practitioners who are not familiar with these concepts to apply the findings directly.


- The paper does not include empirical studies, real-world examples, or relevant demonstrations to validate its theoretical findings, limiting its applicability.

---

> ### Author Response · Authors · 2023-08-07
>
> We thank the reviewer for the valuable and positive comments on our paper and the helpful suggestions for improving it.
> We completely agree with the reviewer that our paper relies on mathematical and theoretical concepts. Using symmetric groups and transposing the ranking evaluation metrics to a heavily mathematical level allows us to overcome the difficulties encountered in a fair evaluation of the metrics in specific contexts (such as rank aggregation methods, Recommender Systems, or Information Theoretical approaches). This is also why we do not include real-world studies but constrain the evaluation to rankings potentially deriving from any of the mentioned algorithms.
> An evaluation for specific cases would hinder the proposed paper's theoretical approach, which should not be seen as guidelines for specific exigencies but as a generalization of ranking evaluation metrics usually applied to specific contexts to more general.
> However, following the comments and suggestions by the reviewer, in the revised version of the paper, we now highlight more clearly the practical consequences resulting from our findings. Specifically, in the Conclusion and Discussion section, we provide a detailed description to clarify for which contexts each property is particularly desirable. By doing this, we aim to provide practitioners looking for specific exigencies with straightforward insights to choose a suitable metric.

---

### Review · Reviewer_jTAd · 2023-07-24

**Summary Of Contributions:**

This paper considers a variety of established metrics for evaluating recommender systems. The authors consider rankings to be members of the symmetric (permutation) group, and apply this mathematical perspective to consider an evaluation metric as a function between two permutations. They proceed to survey different evaluation metrics, measure consistency between pairs of evaluation metrics, and define/survey properties that are desirable for recommendation.

**Audience:**

Yes

**Broader Impact Concerns:**

No concerns about broader impact

**Claims And Evidence:**

No

**Requested Changes:**

### Critical
- Clarify how much of the submission is surveying existing work, and how much is claimed as novel contributions
- Cite related work such as [1], clearly state how this paper complements/expands on related studies [1-2], and directly compare findings where appropriate e.g. Figure 1 and Table 2.
- Resolve conflicting definitions of ground truth, error-based metrics, and symmetry
- Fix typos in the proof of Lemma 6.3 (absolute values)
- Define all relevant terms and metrics (in the Appendix if there are space constraints)

### Non-critical
- Unify "consistent" vs "inconsistent" in Definitions 1 and 2
- Relate properties back to the application of recommender systems

[1] P. Diaconis, Group Representations in Probability and Statistics, Lecture Notes-Monograph Series. Institute of Mathematical Statistics, ISBN 0-940600-14-5, 1988. https://jdc.math.uwo.ca/M9140a-2012-summer/Diaconis.pdf

[2] Valcarce et al. Assessing ranking metrics in top-n recommendation. Information Retrieval Journal, 23:411–448, 2020.


**Strengths And Weaknesses:**

### Strengths
- Describes intuitive properties for comparing different ranking metrics
- Surveys a wide variety of ranking evaluation metrics, which are relevant to the TMLR audience

### Weaknesses
Clarity
- This paper is somewhat unclear in what is being defined for the first time. Are the authors claiming that the group-theoretic perspective is novel? Is this proposing a new mathematical theory, or just studying a new combination of (property, evaluation metric) distinct from what has been established previously? How many properties have been defined before, in the context of recommender systems or otherwise? Some work is cited in related work sections, but it should be discussed throughout the text where appropriate
    - In particular, relating rankings to permutation groups appears as early the 1980s [1]. Chapter 6 of [1] describes distances on permutation groups which correspond to different ranking evaluations (including Spearman's and Kendall's), which is very relevant to Section 6.6 of this submission.
- In general the paper would benefit from being much more self-contained. Many terms are mentioned before their definition (DCG, nDCG, maximal agreement property, Kendall's $\tau$) or entirely without definition (closed set of group elements, cyclic permutation, "sheer" dimensional space, metrics such as Jaccard index and F1 score)
- Some definitions have precise notation (symmetric ranking evaluation metric, mathematical metric) and others are quite informal (Type I Robustness). Some claims have detailed proofs whereas others only have brief, informal proof sketches.
- Definition 1 uses "inconsistent" while Definition 2 uses "consistent"

Correctness
- Propositions 6.1 and 6.2 claim a particular metric is "the only metric" having a certain property (identity of indiscernibles, Type II robustness). It should be made clear that this means "the only metric considered in this paper". Otherwise, the authors would have to prove rigorously that no other evaluation metric could be defined which has the property.
- Ground truth, error-based metrics, and symmetry: These statements and definitions should be reconciled so the paper is consistent - "The error based metrics generally compute the difference between the true and predicted values", "For correlation based and error based metrics, it is easy to prove that they satisfy the symmetry property...", "In conclusion, all metrics involving a ground truth are not symmetric", "All metrics relying on a ground truth ranking can not satisfy the symmetry property"

Organization
- Several properties in Section 6 are proposed without motivation. Often these are desirable intuitively, but the paper would be stronger if it related the properties back to the original recommendation problem. Which properties are most desirable for a particular application? Do some metrics dominate others, or are some better given certain assumptions? One way to accomplish this is to expand on the discussion in Section 8

[1] P. Diaconis, Group Representations in Probability and Statistics, Lecture Notes-Monograph Series. Institute of Mathematical Statistics, ISBN 0-940600-14-5, 1988. https://jdc.math.uwo.ca/M9140a-2012-summer/Diaconis.pdf

---

> ### Author Response · Authors · 2023-08-07
>
> Thank you for the useful suggestions and comments. Following the reviewer's report, we addressed the mentioned points in the revised version of the paper as follows:
>
> * Our contribution to the community considers rankings as elements of symmetric groups and analyzes a comprehensive number of ranking evaluation metrics relying on the mathematical precision of group theory. We define mathematical properties that can be evaluated for each metric independently from the context in which it is used or in which it has been created. We now explicitly state in the Introduction the contributions of the paper.
> * As correctly underlined by the Reviewer, we now explicitly acknowledge that the contribution of Diaconis [1] is highly relevant to the research area. We acknowledge this work in the Introduction, Related Work, Section 6.3, and Appendix A (changes are marked in red). We gave particular attention to explaining to which extent we built up this work and which challenges the paper did not consider.
> * As the Reviewer promptly noticed we did not provide many definitions due to the lack of space and page restriction. We now defined the metrics used in Appendix B and gave insights into them. Following the reviewer's remarks, we also define the previously undefined mathematical terms: swaps (Section 3), closed under group operations (Section 3), cycles (Section 3), and the metrics in the Appendix; we substituted 'sheer' with 'high' -- which fits better in this context. Thank you for the suggestions! We also mention that some mathematical concepts are referred to via different names in the literature (e.g., swaps in Section 3).
> * Again, because of the page limit, we reported in the previous version of the paper essential (in our view) proofs in high detail and only sketches for others. We now add a section of the Appendix in the revised version where we reported longer versions of each shortened proof.
> * Thank you for noticing various typos and kindly reporting them; sorry for the inconvenience. We fixed 'consistent and inconsistent' throughout the text and used mostly 'consistent and non-consistent'; we provided the definitions for each term. We also fixed the misunderstanding about the ground truth labels for error based metrics.
> * Finally, and most importantly, we agree about the need for an extended discussion on the properties, mentioning where they have been introduced in other contexts and why they are important for ranking evaluation metrics. In the revised version of the paper, we now added a new paragraph to the Discussion and Conclusions, mentioning for which application each property is particularly desirable. The aim is to give clarity and practical insights for practitioners to properly choose the most suitable metric for particular research questions and settings.

---

### Review · Reviewer_eqEy · 2023-07-26

**Summary Of Contributions:**

The paper studies a list of evaluation metrics for rankings based on whether they measure sets constructed from these rankings, or are sensitive to order. The authors study a list of axiomatic properties and explain which metric does or does not satisfy these properties.

**Audience:**

No

**Claims And Evidence:**

Yes

**Requested Changes:**

- refocus the presentation in order to clarify what is new, why the definitions are important to the practitioner, and how to use the theoretical results when choosing an evaluation metric
- use more standard definitions (eg for the DCG in proposition 6.1)


**Strengths And Weaknesses:**

1- The presentation is a bit weird, the metrics the authors study have not been introduced for recommender systems, but rather either information retrieval (precision/recall/DCG) or are standard metrics between (partial) rankings (spearman correlation, kendall's tau, etc.) way before recommender systems became a well-identified scientific domain.
2- comparing metrics based on set-based vs ranking-based has some history and I would consider it mostly folklore, see eg https://www.cs.cornell.edu/~caruana/perfs.kdd04.revised.rev1.pdf "Data Mining in Metric Space: An Empirical Analysis of Supervised Learning Performance Criteria" (KDD 2006, Caruana & Niculescu-Mizil) which compares AUC, accuracy, F-score, MSE, "divided the nine metrics into three groups:
threshold metrics, ordering/rank metrics, and probability metrics.". The inconsistency of set-based and ranking metrics is the precise justification of the study of AUC-optimizing algorithms (and more generally algorithms using ranking losses such as pairwise losses) and even among ranking metrics some focus more on "the top of the list" than the midldle/end. The book "Learning to Rank for Information Retrieval" by Tie-Yan Liu (2010) should have exhaustive discussions on these.

3- proposition 6.1 I don't understand where the definition of DCG used by the authors comes from. End of page 6 they use:
DCG(sigma) = sum_{i=1}^n sigma(i)/log(1+i)

The DCG is usually used with ground truth relevance values (let's call it y=(y_1, ..., y_n) with eg y_i \in \{1, ..., 5\} for ratings, and the definition is
DCG(sigma, y) = sum_{i=1}^n y_{sigma(i)}/log(i+1)

which doesn't satisfies the "identity of indiscernible" unless all documents have different relevance values.

4- Proposition 6.2 it is unclear why "MSE, RMSE, MAE, MAPE, R2
score, prevalence, Kendall’s τ score and Spearmann’s ρ are the ***only*** metrics". The property seems to be satisfied by any symmetric

5- overall there are a lot of definitions in this paper and it is unclear to me what they bring to the discussion of choosing a metric in practice. The main message that we need to understand what these metrics try to measure seems well-known to me, and I am not sure what actionable properties we did not know before can be extracted from the paper in its current form. Both the introduction and conclusion claim that it is important "to understand these metrics", but it is unclear what the specific insights of the paper bring in practice (what practical decision will change in light of the results of the paper).

---

> ### Author Response · Authors · 2023-08-07
>
> Thanks for the careful reading and the critical but helpful comments. Following the Reviewer's suggestion, we now explicitly clarified our contributions in the Introduction and Related Work, clearly stating on which work we add up, what is new, and what refers to existing results from the literature. Furthermore, we clarified in the Discussion and Conclusions the need for each of the mentioned properties to be defined and in which contexts they are useful; the paragraph adds up to the mentioned contextualizations already included in the previous version of the paper. For the remaining specific comments, we want to provide a point-by-point reply to the helpful remarks of the Reviewer.
>
> 1. and 5. With our contribution, we aimed to collect a large number of metrics that are used in the literature to compare rankings. Some of them are typical for Recommender Systems and Information Retrieval methods evaluation, while others derive from the accuracy evaluation of classification/regression models; saying this, we do not want to claim that they were first introduced for one particular method or another. All confusion matrix based metrics were well known before any of the mentioned techniques appeared. We claim, though, that they are mostly used in these specific domains within machine learning and computer science.
> 2. We now cited the paper referred by the Reviewer in the Related work section and found them interesting and in line with our work. About Liu et al., we do not completely agree as the book is meant for Recommender Systems/Information Retrieval; However, we aim to obtain a more general review of ranking evaluation metrics, not limited to this single application (where we agree, using top $k$ ranked elements is usually sufficient).
> 3. Following the reviewer's comment, we now added the definition of DCG and nDCG in Appendix C.4; We also explain the choice of this formulation in detail in the Appendix. Adding additional information, such as relevance scores to the elements in the ranking, was not 'fair' with respect to the other ranking evaluation metrics, and considering only binary relevance score was also excluded in order to distinguish the two rankings, i.e., the IoI property was directly not satisfied. Also, distinguished relevance scores are not unrealistic in most contexts.
> 4. As suggested by the reviewer, we added some lines of text to include more explanations for the claim of 'uniqueness' in the proof. We also give here a brief proof of why it is trivial finding counterexamples, for example, for confusion matrix based metrics. It is sufficient to find three rankings $\mu$, $\nu$, and $\tau$ and consider only the first element in the rankings as relevant or retrieved. Then, we aim to prove that $|\{\mu(\tau(1))\}\cap \{\nu(\tau(1))\}| \neq |\{\mu(1)\}\cap \{\nu(1)\}|$; using $\nu = \text{id}$, $\mu = (1\ 3)$ and $\tau =(1\ 2)$, the number of elements in the intersection between relevant and retrieved sets has changed (for arbitrary long rankings). Hence, the confusion matrix on which the confusion matrix based metrics are defined has also changed.

---

### Decision · Action_Editors · 2023-09-08

**Recommendation:** Reject

**Comment:**

Ultimately, all three reviewers leaned towards rejection, with two reject recommendations and one leaning reject.

Several positive aspects of the work were appreciated:
+ Describing intuitive properties for comparing ranking metrics was appreciated (jTAd)
+ Surveying a wide variety of ranking evaluation metrics was appreciated (jTAd)
+ The theoretical approach was considered novel (vkRs), although another reviewer (jTAd) had concerns about ciarity of claimed novelty.
+ The introduced agreement ratio was considered useful. (vkRs)
+ Potential applicability of the framework to informed choice of evaluation metrics was appreciated (vkRs), although another reviewer had doubt about whether they can help choose evaluation metrics, potentially lessening interest in the work (eqEy)

However, the reviewers also raised several concerns:
- Lack of clarity in the definitions and the aimed novelties was criticized (jTAd), in particular which parts are intended as novel.
- There was a wish to address some missing related work and compare findings (jTAd)
- There was concern about the suitability of the studied metrics, and on the other hand relationship to AUC-optimizing algorithms (eqEy)
- It was unclear whether the various propositions would help choose metrics in practice (eqEy)
- There was a wish to make the paper more self-contained in terms of providing definitions before they are used. (jTAd)
- Lack of detail in some notations and proofs was criticized (jTAd)
- There was lack of clarity in some of the propositions (eqEy)
- There was criticism of propositions 6.1, 6.2 in terms of setting their scope properly (jTAd), and for several definitions better consistency was desired (jTAd)
- Missing motivations for properties in Section 6 were also criticized. (jTAd)
- Lack of empirical studies, real-world examples, or demonstrations was criticized (vkRs)

Although authors provided rebuttals, ultimately several of the concerns remained. Thus it seems the work is not ready for publication in TMLR.

**Audience:**

If the novelty had been properly clarified and sufficient clarity improvements in other parts were made, and if suitable studies were included as evidence for the author's claim that the work "allows for a conscious choice of evaluation metrics to measure the similarity among rankings in specific contexts", then the work could certainly have an audience among TMLR readers. In its current state, some individuals might still be able to find use for the results.

**Claims And Evidence:**

The reviewers raised several concerns especially regarding clarity in key aspects (lack of clarity in which parts were intended as novel; lack of clarity in how the definitions would affect choosing a metric in practice, and lack of clarity along various technical aspects). Moreover, lack of empirical studies, real-world examples, or demonstrations was criticized. Thus, overall, it seems the claims are not supported by convincing evidence, and are not clear in some details.

Although authors claimed the work "provides a theoretical framework and we detach from specific contexts of application", several statements still argue for the interestingness of the work based on helping choose a metric: the abstract motivates by "challenges of finding the right evaluation metric" and "choosing one metric despite another", and state that the findings allow "a more conscious choice based on specific exigencies" - the intro similarly states that "Choosing evaluation metrics to compare two rankings is often non-straightforward" and states about their analysis "Eventually, this allows for a conscious choice of evaluation metrics". So authors do still try to motivate by a connection to helping people choose metrics. Moreover, in the conclusions, they end by saying "Having collected the obtained theoretical and experimental results in a concise table, we allow for insights of immediate use" which also may be to trying to argue for applicability.

Authors should tone down some of the claims of "immediate use" and further clarify the theoretical nature of the work and that any extensive empirical evaluation is left for future investigation; or if claims of immediate use are desired authors should include empirical studies to justify them.